# Transcriptome-wide association analyses reveal the impact of regulatory variants on rice panicle architecture and causal gene regulatory networks

Luchang Ming [1,6], Debao Fu[1,6], Zhaona Wu[1], Hu Zhao[1], Xingbing Xu[1], Tingting Xu[1], Xiaohu Xiong[1], Mu Li[1], Yi Zheng[1], Ge Li[1], Ling Yang[1], Chunjiao Xia[1], Rongfang Zhou[1], Keyan Liao[1], Qian Yu[1], Wenqi Chai[1], Sijia Li[1], Yinmeng Liu[1], Xiaokun Wu[1], Jianquan Mao[1], Julong Wei[2], Xu Li[1], Lei Wang[1], Changyin Wu [1] ✉ & Weibo Xie [1,3,4,5] ✉

Panicle architecture is a key determinant of rice grain yield and is mainly determined at the 1-2 mm young panicle stage. Here, we investigated the transcriptome of the 1-2 mm young panicles from 275 rice varieties and identified thousands of genes whose expression levels were associated with panicle traits. Multimodel association studies suggested that many small-effect genetic loci determine spikelet per panicle (SPP) by regulating the expression of genes associated with panicle traits. We found that alleles at *cis*-expression quantitative trait loci of SPP-associated genes underwent positive selection, with a strong preference for alleles increasing SPP. We further developed a method that integrates the associations of *cis*- and *trans*-expression components of genes with traits to identify causal genes at even small-effect loci and construct regulatory networks. We identified 36 putative causal genes of SPP, including *SDT (MIR156j)* and *OsMADS17*, and inferred that *OsMADS17* regulates *SDT* expression, which was experimentally validated. Our study reveals the impact of regulatory variants on rice panicle architecture and provides new insights into the gene regulatory networks of panicle traits.

Rice (*Oryza sativa*) is an important food crop that feeds billions of people worldwide. Panicle architecture is a key factor in determining rice grain yield, therefore identifying genes involved in panicle development, exploring their natural variation, and understanding their regulatory mechanisms are essential for breeding high-yielding rice varieties[1]. Based on mutant analyses or bi-parental populations, scientists have cloned several genes involved in panicle development, such as *OSH1*[2], *LAX1*[3], *Gn1a*[4], *OsSPL14*[5,6], *DEP1*[7], and *FZP*[8]. However, their

[1]National Key Laboratory of Crop Genetic Improvement, Hubei Hongshan Laboratory, Huazhong Agricultural University, Wuhan, China. [2]Center for Molecular Medicine and Genetics, Wayne State University School of Medicine, Detroit, MI, USA. [3]Shenzhen Institute of Nutrition and Health, Huazhong Agricultural University, Wuhan, China. [4]Shenzhen Branch, Guangdong Laboratory for Lingnan Modern Agriculture, Genome Analysis Laboratory of the Ministry of Agriculture, Agricultural Genomics Institute at Shenzhen, Chinese Academy of Agricultural Sciences, Shenzhen, China. [5]Hubei Key Laboratory of Agricultural Bioinformatics, College of Informatics, Huazhong Agricultural University, Wuhan, China. [6]These authors contributed equally: Luchang Ming, Debao Fu. ✉ e-mail: cywu@mail.hzau.edu.cn; weibo.xie@mail.hzau.edu.cn

**Fig. 1 | Overview of the study.**

regulatory networks are not fully understood, and for most of them, the favorable alleles remain unexplored[9]. In recent years, genome-wide association studies (GWAS) have been widely used in plants to explore the abundant natural variations, and GWAS for panicle traits have been reported in rice[10–13]. However, only a few genes under quantitative trait loci (QTLs) have been identified by GWAS as the large linkage disequilibrium (LD) decay distance of the rice genome and the complexity of the genetic basis of panicle traits. Therefore, it is necessary to utilize advanced genomic technologies and develop appropriate methods for a more comprehensive and effective understanding of the regulatory mechanisms of rice panicle traits.

Gene expression variation regulated by non-coding *cis*-variants or variations of *trans*-factors is a significant contributor to phenotypic diversity among varieties[14,15]. In rice, several genes such as *Gn1a*[4], *IPA1*[16], and *FZP*[17] have been reported to have *cis*-variants that regulate gene transcription, leading to variation in panicle architecture. In humans, the Genotype-Tissue Expression (GTEx) project collects and analyzes transcriptome data of multiple tissues from different individuals, demonstrating the effectiveness of population transcriptome data in identifying genetic variations that can explain differences in gene expression among individuals, known as expression quantitative trait loci (eQTLs). Currently, we are running a Rice GTEx project that aims to build a comprehensive resource to study tissue-specific gene expression and regulation in rice. This will help identify causal genes and understand the molecular processes through which genetic variations affect agronomic traits.

This study is part of the Rice GTEx project and focuses on developing analysis strategies appropriate for crops and understanding gene regulation related to rice panicle traits. In this study, we collected young panicle samples and conducted a transcriptomic study on 275 representative rice varieties (Fig. 1). We identified thousands of genes whose expression levels are associated with panicle traits. We explored the role of selection on gene expression. We further developed a method more appropriate for crop studies to identify causal genes at even small-effect GWAS loci and to construct regulatory networks. Finally, we demonstrated the effectiveness of this method by validating *OsMADS17* as a causal gene that affects the number of spikelets per panicle (SPP) and regulates the expression of another causal gene, *SDT*. Our study provides a valuable resource for understanding the regulatory mechanisms of rice panicle traits and potential targets for molecular breeding.

## Results

### Genome-wide association studies reveal the complex genetic structure of panicle traits

We conducted four field experiments across geographic regions and years using a population of 529 representative rice varieties[18] and collected panicle traits including SPP, number of primary branches (NPB), and panicle length (PL) (Supplementary Data 1). We assessed the consistency of the panicle traits and found that these traits were highly consistent across locations and years, with Pearson's correlation coefficients (PCC) ranging from 0.45 to 0.85 (Supplementary Fig. 1a–d). To yield more robust association results, we used the best linear unbiased prediction (BLUP) model to combine phenotypes of the four experiments, and then performed GWAS on the BLUP value of panicle traits of 529 varieties. At a genome-wide significance threshold of $2.54 \times 10^{-8}$, only one, three, and eight significant QTLs were identified for SPP, NPB, and PL, respectively (Fig. 2a–c). However, there are many apparent peaks at the threshold of $10^{-5}$ (Supplementary Data 2). This is similar to the GWAS results of SPP in a previous study using this population[12], or studies using 950 rice varieties[11] or 1495 hybrid rice varieties[19]; that is, there are only a few genome-wide significant loci, but there are more obvious peaks below the threshold line.

We estimated the heritability of the three traits based on genetic variants using GCTA[20] and found that all three traits had high heritability (0.852, 0.820, and 0.858 for SPP, NPB, and PL, respectively). This result, combined with the results of association analysis of SPP and NPB (lack of significant major effect loci but many peaks below the significance threshold), suggests that SPP and NPB may be more regulated by small-effect loci. To confirm this speculation, we used LASSO[21] to estimate the phenotypic variances explained by GWAS loci identified at different thresholds ("Methods"). We found that the variances of SPP and NPB explained by GWAS loci were relatively small at the threshold of $10^{-7}$ (estimated by 10-fold cross-validation $R^2$ between the predicted and the observed phenotypes, hereafter $R_{cv}^2$; 0.205, 0.161, and 0.419 for SPP, NPB, and PL, respectively; Fig. 2d–f). As the GWAS threshold relaxed, the explained variances of SPP and NPB increased (0.880, 0.876, and 0.868 for SPP, NPB, and PL, respectively, at the threshold of $10^{-3}$). Further permutation tests confirmed that the phenotypic variances explained by the small-effect loci of GWAS were significant (Supplementary Fig. 2). We also performed GWAS on 275 varieties for which transcriptome data were acquired and obtained

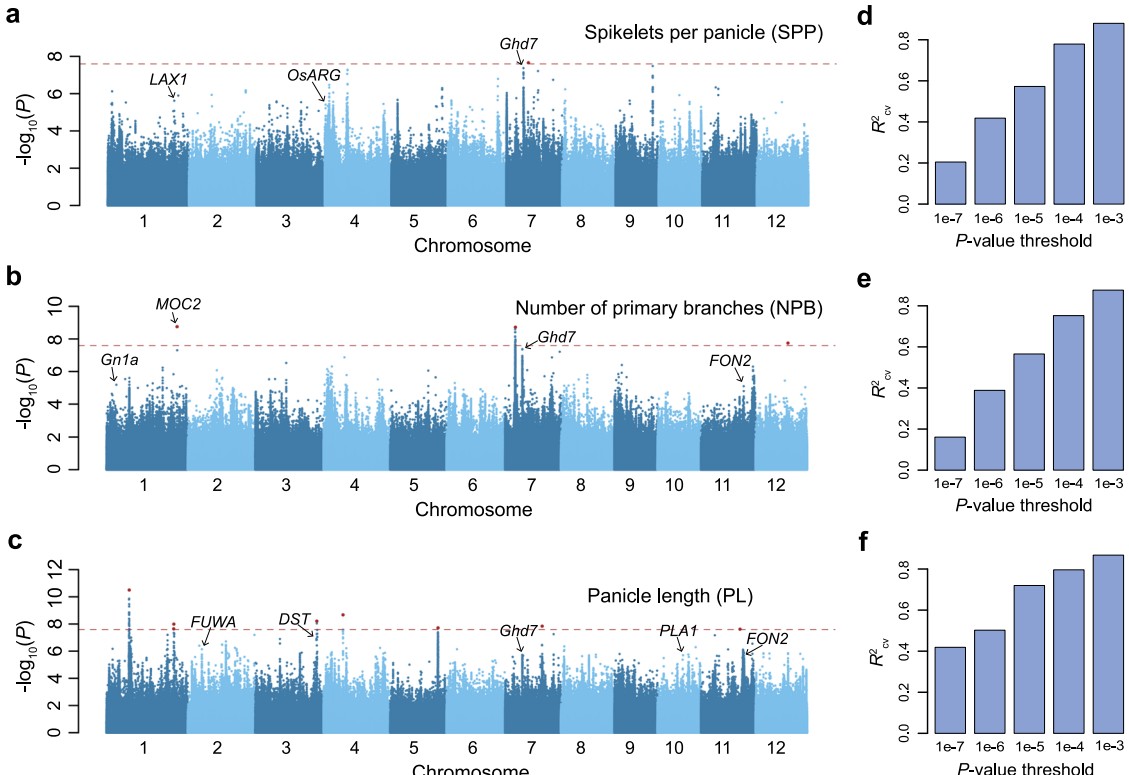

**Fig. 2 | GWAS and genetic variance composition of panicle traits. a–c** Manhattan plots of GWAS using linear mixed model for the number of spikelets per panicle (SPP), the number of primary branches (NPB), and panicle length (PL). The red dashed lines indicate the genome-wide significance threshold ($2.54 \times 10^{-8}$, multiple comparisons corrected by 0.05/No. independent variants), and the red points indicate lead variants of genome-wide significant QTLs. Genes that have been reported to be related to panicle architecture within 100 kb of significant loci at the $1 \times 10^{-5}$ threshold are marked out. **d–f** The phenotypic variance explained by GWAS lead variants identified at different $p$ value thresholds for SPP, NPB, and PL. The explained phenotypic variances are assessed by tenfold cross-validation using LASSO. $R^2$: predicted values versus observed phenotypes.

similar results (Supplementary Figs. 3 and 4). These results suggested that variations in SPP and NPB in rice varieties are mainly regulated by numerous small-effect loci.

**Transcriptome-wide association studies reveal key genes associated with panicle traits**

To further explore the regulatory mechanisms of panicle traits and characterize the variability of the transcriptomes in young panicles of different varieties, we generated the transcriptome data for 275 rice varieties with 1–2 mm young panicles at the branch primordia differentiation stage (Supplementary Data 3), which determines the NPB and further affects the number of SPP[22]. Consistent differences at RNA levels and genomic levels were observed among the varieties (Supplementary Fig. 5a–e), indicating that the variability of the transcriptomes are mainly caused by genetic variability and also indicating the high quality of our data. A total of 30,869 genes (including microRNA genes) were considered as expressed and used for further analysis ("Methods").

We carried out transcriptome-wide association studies (TWASs) by associating the expression values of each expressed gene in the population with panicle traits. At a false discovery rate (FDR) of 0.05, 4175, 5844, and 6839 significantly associated genes were detected for SPP, NPB, and PL, respectively (Supplementary Data 4–6). Of the 20 genes most significantly associated with SPP, 10 are transcription factors, including two YABBY transcription factors and six MADS transcription factors (Supplementary Data 4). Of these, *OsSh1* ranked first ($p$ value = $7.31 \times 10^{-14}$) and is a YABBY transcription factor reported to control seed shattering and undergo parallel selection during domestication in multiple cereals[23]. Another YABBY is *OsYABBY1* ($p$ value = $1.76 \times 10^{-10}$, rank = 17) which is specifically expressed in the palea and lemma from their inception and controls spikelets development[24]. The six MADS transcription factors include *OsMADS1* ($p$ value = $6.85 \times 10^{-12}$, rank = 8), *OsMADS7* ($p$ value = $2.32 \times 10^{-11}$, rank = 12), *OsMADS8* ($p$ value = $5.88 \times 10^{-12}$, rank = 6), *OsMADS3* ($p$ value = $1.49 \times 10^{-12}$, rank = 4), and two *AGAMOUS-LIKE6* genes (*OsMADS6*, $p$ value = $1.51 \times 10^{-10}$, rank = 16; *OsMADS17*, $p$ value = $1.77 \times 10^{-10}$, rank = 18). These MADS genes have been found to be important in controlling spikelet initiation and spikelet organ development[25–31]. Our results show that the expression of these genes is negatively correlated with SPP, which is also consistent with the view that inhibiting the transformation of spikelets prolongs the development time of branches and increases SPP[32]. These results indicate the important value of our data in uncovering the regulatory direction of genes for panicle traits.

We further surveyed the known panicle development-related genes among the top 500 most significantly associated genes and found *OsSPL14* ($p$ value = $4.06 \times 10^{-8}$, rank = 87), *OSH1* ($p$ value = $8.59 \times 10^{-8}$, rank = 104), *OsPID* ($p$ value = $7.26 \times 10^{-6}$, rank = 405), and *PLA1* ($p$ value = $5.38 \times 10^{-7}$, rank = 184) were in the list (Fig. 3a and Supplementary Fig. 6a, b). The ideal plant architecture gene *IPA1*, which encodes *OsSPL14*, has been found to have a larger panicle size and increased SPP under high expression conditions[6,16]. We found that the expression level of *IPA1* was positively correlated with SPP. *IPA1* is under the regulation of *miR156* and *miR529*, so the up-regulation of *miR156* and *miR529* inhibits the function of *IPA1*, which causes the panicle to be smaller[33]. Our results are consistent with this, where the expression levels of *SDT*[34] (*MIR156j*) and *MIR529a* are negatively associated with *IPA1* (PCC = −0.32 and −0.42 for *MIR156* and *MIR529a*, respectively) and SPP ($p$ value = $5.72 \times 10^{-7}$ and $1.50 \times 10^{-12}$, rank = 185 and 5 for *MIR156j* and *MIR529a*, respectively). In addition, we also

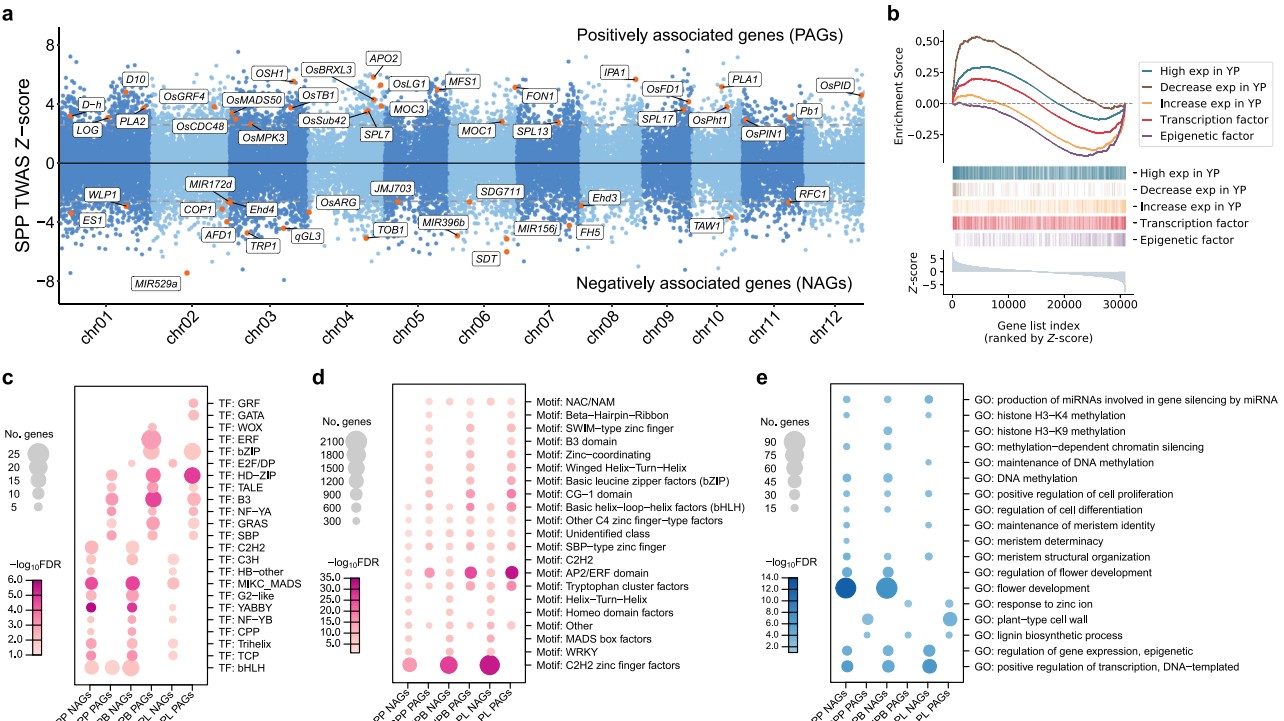

**Fig. 3 | Transcriptome-wide association study (TWAS) for SPP and enrichment analysis of TWAS significant genes. a** Manhattan plot of TWAS for SPP. TWAS $Z$-scores ($y$-axis) are plotted against gene positions ($x$-axis) on each of the chromosomes. The gray dashed lines indicate the $Z$-scores at the FDR of 0.05, and genes exceeding this threshold are defined as TWAS significant genes in the following analysis. Known genes related to panicle development are labeled and marked as orange dots. **b** Gene Set Enrichment Analysis for TWAS significant genes of SPP. All genes were sorted in descending order ($x$-axis) by the TWAS $Z$-scores. The $y$-axis of the bottom panel indicates the TWAS $Z$-scores. Genes from different gene sets are shown in different colors. In the middle panel, a gene belonging to a certain gene set is indicated by a colored vertical line. In the top panel, the enrichment scores[36] ($y$-axis) for each gene set are plotted against the rank of TWAS $Z$-scores ($x$-axis). High exp in YP: genes preferentially expressed in young panicles. Decrease exp in YP: genes progressively down-regulated in panicle development. Increase exp in YP: genes progressively up-regulated in panicle development. **c–e** Transcription factor families (**c**), TF binding motifs (**d**), and gene ontology terms (**e**) enriched in TWAS significant genes for panicle traits (FDR <0.05). NAGs and PAGs: negatively and positively associated genes, respectively. The bubble size indicates the number of overlapping genes and the bubble color indicates the −log$_{10}$ FDR of the enrichment.

identified other miRNAs associated with panicle traits, such as *miR396*[35] ($p$ value = $1.61 \times 10^{-6}$, rank = 258 for SPP) and *miR172*[33] ($p$ value = $6.07 \times 10^{-5}$, rank = 1092 for NPB) which have been reported to play an important role in panicle development.

We characterized the genes significantly associated with panicle traits. We found that the positively associated genes (PAGs) of SPP enriched for genes that were preferentially expressed in young panicles ($p$ value = $2.39 \times 10^{-4}$) and genes that were progressively down-regulated in panicle development ($p$ value = $3.31 \times 10^{-4}$), while the negatively associated genes (NAGs) enriched for genes that were progressively up-regulated in panicle development ($p$ value = $6.06 \times 10^{-4}$, Fig. 3b, Supplementary Fig. 6e and Supplementary Data 7). GO enrichment analysis showed that NAGs of SPP were enriched for functions such as flower development, meristem structural organization, and meristem determinacy, as well as for transcriptional regulation, DNA methylation, histone methylation, and miRNA production (Fig. 3e and Supplementary Data 8). These results suggest that TWAS detected a large number of regulatory factors that may be important for SPP and that the NAGs play a vital role during panicle development. To further confirm that TWAS significant genes contain many regulatory factors, we performed Gene Set Enrichment Analysis (GSEA)[36]. We found that the NAGs of SPP were significantly enriched for epigenetic factors and transcription factors (Fig. 3b and Supplementary Data 7). Specifically, PAGs were enriched for transcription factors from families such as B3, SBP, and NF-YA, and NAGs were enriched for transcription factors from families such as MADS, YABBY, TCP, and C2H2 (Fig. 3c and Supplementary Data 9), suggesting that TWAS significant genes from these families may play important

roles in panicle development. We further performed motif enrichment analysis on the promoter sequences of TWAS significant genes using Plant Regulomics[37]. We found that the NAGs of SPP were enriched in motifs of families such as C2H2 zinc finger, MADS, and WRKY (Fig. 3d and Supplementary Data 10), while PAGs of SPP were enriched in motifs of families such as B3 and bZIP, while motifs of SBP and AP2 families were enriched in the regulatory sequences of both PAGs and NAGs of SPP. The enrichment of B3, SBP, MADS, and C2H2 family transcription factors in the TWAS significant genes of SPP and NPB, as well as the enrichment of their binding motifs in the promoter sequences of the TWAS significant genes, implies an underlying regulatory network of these transcription factors with downstream genes. In conclusion, the results of TWAS suggest that our data are valuable in uncovering panicle trait-related genes and their regulatory networks.

## A large number of TWAS-significant genes are regulated by numerous small-effect loci

When examining the expression of TWAS significant genes for panicle traits in different varieties by heat map, we found that many genes have similar expression patterns (Fig. 4a and Supplementary Fig. 7), implying that these genes might be regulated by the same genetic loci (i.e., eQTL hotspots) to some extent. We were particularly interested in whether there are GWAS loci (pQTLs) that play a role in regulating the expression of multiple TWAS significant genes, namely pQTL-eQTL hotspots. We first associated TWAS significant genes (FDR <0.05) with lead variants of pQTLs identified using the 529 varieties at the suggestive threshold ($p$ value = $5.07 \times 10^{-7}$). However, for the SPP and NPB, no pQTL were found to be associated with more than 10 TWAS

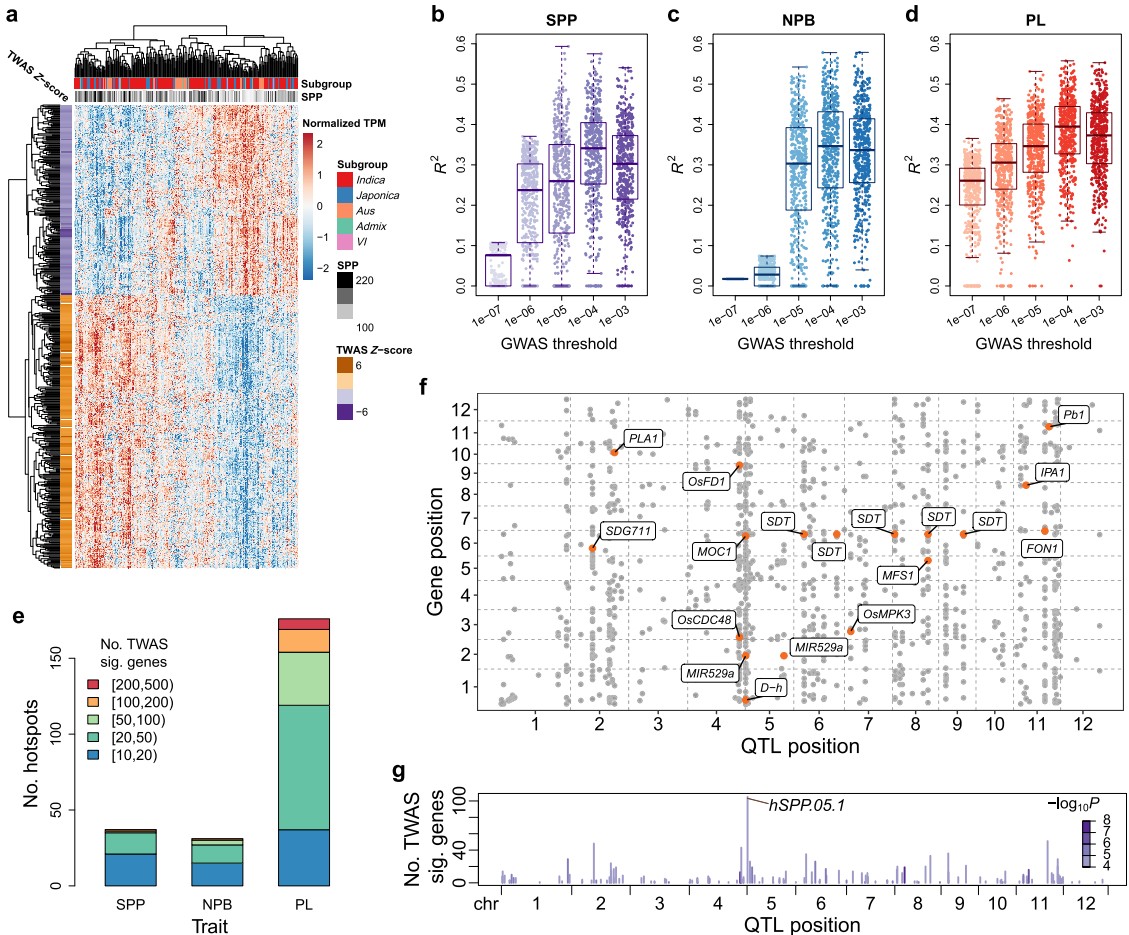

**Fig. 4 | Associations between GWAS QTLs and TWAS significant genes.**
**a** Expression patterns of the top 500 TWAS significant genes of SPP in the 275 varieties. Each column is a variety, and each row is a gene. **b–d** $R^2$: the square of the Pearson correlation coefficient between the panicle trait and the predicted expression values of the top 500 TWAS significant genes of SPP (**b**), NPB (**c**), and PL (**d**), respectively. Each point represents a gene. The predicted expression values are predicted using the LASSO model based on lead variants of GWAS QTLs identified at different thresholds in the 275 varieties' panel. For each box plot, the center line represents the median, the box's lower and upper boundaries indicate the first and third quartiles, and the whiskers extend to data points within 1.5 times the interquartile range from the box. **e** Bar plot of the number of pQTL-eQTL hotspots for TWAS significant genes. A GWAS QTL of panicle traits (pQTL, $p$ value $<1\times10^{-4}$) is defined as a pQTL-eQTL hotspot if it is an expression quantitative trait locus (eQTL,

$p$ value $<1\times10^{-4}$) for many genes and these genes are significantly enriched for more than 10 TWAS significant genes (BH-adjusted one-sided Fisher's exact test $p$ value $<0.05$). The pQTL-eQTL hotspots are categorized according to the number of TWAS significant genes in the associated targets and indicated by different colors. **f** Associations between the expression levels of TWAS significant genes and pQTLs for SPP. The genomic positions of TWAS significant genes (y-axis) are plotted against the positions of lead variants of pQTLs (x-axis) for each significant association. Known panicle development-related genes associated with pQTL-eQTL hotspots are labeled and marked as orange dots. **g** The number of TWAS significant genes associated with each GWAS QTL of SPP. The color of each bar represents the $-\log_{10} p$ value of GWAS for that QTL. Source data underlying (**b–g**) are provided as a Source Data file.

significant genes at the suggestive threshold, and pQTLs obtained using the 275 varieties were similar, implying that the expression variation of TWAS significant genes might not be regulated by the GWAS-significant loci, which is different from the results reported in other crops[38,39]. On the other hand, genetic correlation analysis suggests that a large part of the correlation between TWAS significant genes and panicle traits could be explained by genetic causes ("Methods", Supplementary Fig. 8a–f). Considering that SPP and NPB are mainly regulated by small-effect loci from our GWAS results, we hypothesized that the expression variation of TWAS significant genes affecting the panicle traits might also be mainly contributed by small-effect loci.

To confirm that speculation, we used LASSO to predict the expression of the top 500 TWAS significant genes in different varieties based on lead variants of pQTLs identified at different thresholds and then calculated Pearson's correlation coefficients between the predicted gene expression values and panicle traits ("Methods"). We found that the correlations were low at the pQTL threshold of $10^{-7}$. However, when the pQTL threshold was relaxed, the correlations

increased substantially (Fig. 4b–d and Supplementary Fig. 8g–i). Further permutation tests showed that the correlations between panicle traits and the predicted gene expression values were significant ($p$ value $<0.002$) at the relaxed pQTL thresholds ($10^{-5}$, $10^{-4}$, and $10^{-3}$; Supplementary Figs. 9 and 10). These results suggest that a large proportion of the variation in the expression of TWAS-significant genes is regulated by small-effect loci. In addition, we noted that the correlations were higher for the 275 accession panel than for the 529 accession panel (for SPP, the highest mean $R^2$ was 0.188 in the 529 varieties and 0.320 in the 275 varieties). It could be that there is still some heterogeneity between the 529 varieties and the 275 varieties, although the 275 varieties were selected as representative of the 529 varieties. Therefore, we used the GWAS results of the 275 varieties panel if not specifically stated in the later analysis.

Since the effect of genetic loci affecting phenotypes as well as TWAS significant genes may be relatively small, we attempted to identify pQTL-eQTL hotspots under a more relaxed threshold ($p$ value $<10^{-4}$ for both pQTLs and eQTLs). A pQTL is defined as a pQTL-eQTL

a
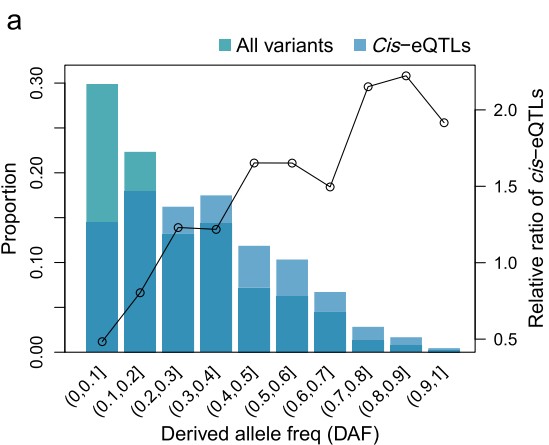

b
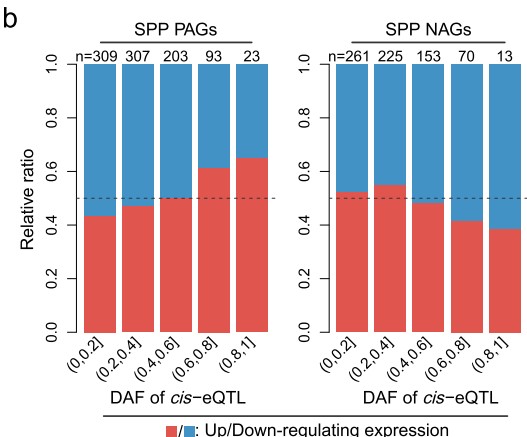

**Fig. 5 | Distribution of derived allele frequency (DAF) at *cis*-eQTLs and impact of derived alleles on the expression of SPP TWAS-significant genes. a** Variants with high DAF are enriched for *cis*-regulatory variants. All variants and lead variants of *cis*-eQTLs were divided into different DAF intervals. The bar plot represents the proportion of all variants (cyan) or lead variants of *cis*-eQTLs (sky blue) in each DAF interval (left *y*-axis). The line indicates the relative ratio of the proportion of *cis*-eQTLs to the proportion of all variants in each DAF interval (right *y*-axis). A total of 13,877 lead variants were used in this analysis. Derived alleles were identified based on wild rice data ("Methods"). **b** High-frequency derived alleles at *cis*-eQTLs of SPP TWAS-significant genes tend to have a positive effect on SPP. The *cis*-eQTLs of SPP PAGs (left panel) and NAGs (right panel) are divided into different intervals (*x*-axis) according to DAF, and the *y*-axis represents the proportion of *cis*-eQTLs whose derived allele is up-regulating gene expression (red) or down-regulating gene expression (blue). The numbers at the top of the bars indicate the number of TWAS significant genes containing *cis*-eQTL in each interval. Source data are provided as a Source Data file.

hotspot if it is an eQTL for many genes and these genes are significantly enriched for more than 10 TWAS significant genes (BH-adjusted Fisher's exact test *p* value <0.05). We identified 37, 31, and 176 pQTL-eQTL hotspots for the TWAS significant genes of SPP, NPB, and PL, respectively (Fig. 4e and Supplementary Data 11), revealing a complex regulatory network for the panicle traits. The regulatory networks of hotspots and target genes reveal possible important regulatory mechanisms of panicle development. For example, one of the most significant hotspots of SPP, hSPP.05.1, was found to be associated with 105 TWAS significant genes, including several known panicle development-related genes, such as *MIR529a*, *MOC1*, and *OsMADS1* (Fig. 4f, g and Supplementary Fig. 11a–d). Another example, hNPB.09.1, was found to be associated with 75 NPB TWAS-significant genes, including several known panicle development-related genes, such as *Ghd2*, *OsNF-YA4*, and *OsGRF10* (Supplementary Fig. 11e–h). However, the regulatory genes in these hotspots remain to be characterized.

**Differential selection on derived alleles impacts gene expression and panicle traits**

Many studies have shown that gene expression is largely regulated by *cis*-regulatory variants[38,40], but whether *cis*-regulatory variants are under selection during domestication or breeding, and how the selection affects agronomic traits in crops has rarely been studied[41]. To assess whether *cis*-regulatory variants tended to be under selection, we identified derived alleles based on allele frequencies in wild rice[42] and analyzed the derived allele frequencies (DAF) of lead variants of *cis*-eQTLs. We observed that as DAF increased, the proportion of variants belonging to lead variants of *cis*-eQTLs gradually increased, suggesting that *cis*-regulatory variants play an important role as targets of natural or artificial selection in rice (Fig. 5a).

We further wondered how the selection for *cis*-regulatory variants affects agronomic traits. We first examined whether the derived alleles of *cis*-eQTLs for the TWAS significant genes also tended to be under selection, and the results showed that the *cis*-eQTLs for the TWAS significant genes had the same preference for high DAF (Supplementary Fig. 12a–c). We then divided TWAS significant genes according to the direction of association with the panicle traits and analyzed the effects of the derived allele of *cis*-eQTL on gene expression (Fig. 5b and Supplementary Fig. 12d, e). We found that the derived alleles with high-frequency preferred to up-regulate the expression of PAGs or

down-regulate the expression of NAGs, i.e., tended to have a positive effect on the phenotype. This result suggests that during rice domestication or breeding, derived alleles that improve panicle traits are positively selected, resulting in higher frequencies, while derived alleles that have negative effects on traits are negatively selected, leading to lower frequencies.

Could such differences in selection be observed in the GWAS signal of panicle traits? We assessed the GWAS signal distribution of *cis*-eQTLs of PAGs and NAGs. Quantile-quantile plots showed that for genes whose derived alleles of *cis*-eQTLs have a positive effect on SPP (up-regulate PAGs or down-regulate NAGs), GWAS signals of *cis*-eQTLs for genes with higher *cis*-eQTL DAF had a greater departure from an expected *p* value distribution (Supplementary Fig. 12f); for genes whose derived allele of *cis*-eQTLs may have a negative effect on SPP (down-regulate PAGs or up-regulate NAGs), GWAS signals of *cis*-eQTLs for genes with lower *cis*-eQTL DAF had a greater departure from the null distribution (Supplementary Fig. 12i). These results suggest that the selected *cis*-eQTLs of TWAS significant genes are enriched for variants that have effects on panicle traits.

Multiple genes that may play important roles in panicle development showed selection on *cis*-regulatory variants. For example, a gene encoding a B3 transcription factor (LOC_Os06g42630), which is positively associated with SPP (Supplementary Fig. 13a), is specifically expressed in young panicles[43] and thus may play a role in panicle development. The derived allele of the *cis*-eQTL (vg0625635377) for this gene up-regulates the gene expression (Supplementary Fig. 13b) and thus may have a positive effect on SPP. The DAF of the *cis*-eQTL is 0.081 in wild rice and raised to 0.799 in cultivated rice. Another example is that a gene containing the DUF567 domain (LOC_Os05g40630), the expression of this gene is negatively associated with SPP (Supplementary Fig. 13d). The derived allele of the *cis*-eQTL (vg0523822557) down-regulates the expression of this gene (Supplementary Fig. 13e), thus the derived allele may have a positive effect on SPP. Further evidence is provided by the *cis*-eQTL and the variants in LD with *cis*-eQTL which show a regional significant association with SPP (Supplementary Fig. 13f). The DAF of this *cis*-eQTL is 0.08 in wild rice and raised to 0.908 in cultivated rice. These results provide insights and clues for studying the selection and effects of domestication or breeding on gene expression and phenotype.

## Decomposition of gene expression identifies causal genes at small-effect GWAS loci for panicle traits

GWAS can rapidly locate QTLs affecting phenotypes, but further identification of candidate genes within the QTL regions remains a challenge. This problem is particularly serious in rice due to the large LD decay distance. Based on the population transcriptome data, we proposed a method to combine information on *cis*- and *trans*-components of gene expression to identify causal genes. As shown in the model in Fig. 6a, the variation of gene expression caused by *cis*-regulatory variants is defined as the *cis*-expression component (*cis*-EC), while the variation of gene expression caused by distal variants is defined as the *trans*-expression component (*trans*-EC). If Gene B is expressed variably among individuals and its *cis*-EC is associated with variation in a phenotype, then Gene B may be a causal gene affecting the phenotype, but it may also be a false positive simply due to LD between *cis*-regulatory variants of Gene B and a nearby causal variant of the phenotype. In contrast, if not only the *cis*-EC of Gene C is associated with the phenotype, but also the *trans*-EC of Gene C is associated with the phenotype, i.e., Gene C may be in a regulatory network that affects the phenotype, and not only its *cis*-regulatory variants may affect the phenotype, but it is also regulated by other regulators (e.g., Gene A in the model) affecting the phenotype, then Gene C is more likely to be a causal gene compared to Gene B.

Based on this idea, we first estimated *cis*-EC of each gene using GCTA based on variants within 100 kb of the gene, and then subtracted *cis*-EC from the gene expression and used the residuals as *trans*-EC (Methods). At the threshold of $1.62 \times 10^{-6}$ (Bonferroni corrected *p* value <0.05), 14,392 genes with significant *cis*-genetic variance were identified and used for subsequent analyses. We then associated *cis*- and *trans*-ECs with panicle traits separately (defined as *cis*- and *trans*-expression component-based association study, abbreviated as *cis*- and *trans*-ECAS) and prioritized genes with both *cis*- and *trans*-EC associated with phenotypes as candidate genes. Consistent with the aforementioned results that panicle traits were primarily regulated by small-effect loci, we found only two genes whose *cis*-EC was significantly associated with SPP and none for NPB (FDR <0.05). At a relatively relaxed threshold (*p* value <0.01), we identified 288, 259, and 477 genes with *cis*-EC associated with SPP, NPB, and PL, respectively. Finally, by considering the results of *trans*-ECAS simultaneously (FDR < 0.01), we identified 36, 48 and 99 putative causal genes (referred to as cis- and trans-ECAS genes) significantly associated with SPP, NPB and PL, respectively (Fig. 6b and Supplementary Data 12).

Some of these candidate genes have been reported to be involved in panicle development. For example, *SDT*[34] encodes a small RNA *MIR156j*, and studies have reported that *miR156* regulates panicle size by down-regulating *IPA1*[5,6]. We found that both *cis*- and *trans*-EC of *SDT* were negatively associated with SPP (*p* value = $4.39 \times 10^{-6}$ for *cis*-ECAS and $1.06 \times 10^{-6}$ for *trans*-ECAS; Fig. 6c), indicating that *SDT* is a causal gene that affects the phenotype of SPP in rice varieties. Using GWAS, we detected variants around *SDT* that are associated with both *SDT* expression and SPP (Fig. 6d), implying that these variants may affect SPP by altering the expression of *SDT*. We further analyzed the natural variation of *SDT*. We first examined the primary transcript of *MIR156j* and found no variants. We then focused on the *cis*-regulatory variants of *SDT* and found that only one SNP (SDT-V1: vg0626550198) located within 5 kb upstream of the transcription start site (TSS) was significantly associated with the expression of *SDT*. Besides, we found a variant (vg0626555871) in the second intron of *SDT*, which is in LD with SDT-V1 but less significantly associated with *SDT* expression and SPP. By checking 33 high-quality rice genome sequences[44], we found that this variant actually is an ~2.7 kb insertion rich in "CTAT" repeats. Moreover, we noticed that two SNPs (SDT-V2: vg0626530751, SDT-V3: vg0626534166) at ~24 kb and 20 kb upstream of the TSS were significantly associated with *SDT* expression but not in LD with SDT-V1. Since SDT-V2 was in LD with SDT-V3 and more significantly associated

with *SDT* expression than SDT-V3, we finally prioritized SDT-V2 as well as SDT-V1 as candidate causal variants. The three haplotypes formed by SDT-V1 and SDT-V2 showed significant differences in *cis*-EC of *SDT* and SPP (Fig. 6e and Supplementary Fig. 15a, b), suggesting a coordinated effect of the two variants on gene expression and the phenotype. Although the association *p* value between the lead variant SDT-V1 and SPP is only $3.53 \times 10^{-5}$, by integrating the association of *cis*- and *trans*-EC with SPP, we were able to identify *SDT* as a causal gene. This demonstrates the effectiveness of our method in identifying causal genes at small-effect loci that affect the phenotype due to *cis*-regulatory variants.

*OsMADS17* is an AGAMOUS-LIKE6 (AGL6) MADS-box gene, and its paralog *OsMADS6* was found to play an important role in regulating rice floral organ identity and floral meristem determinacy[29,45], while their function on SPP has not been reported. We found that both the *cis*- and *trans*-EC of *OsMADS17* were negatively associated with SPP (*p* value = $1.61 \times 10^{-3}$ for *cis*-ECAS and $5.46 \times 10^{-10}$ for *trans*-ECAS, Fig. 7a), suggesting that *OsMADS17* may be a negative regulator of SPP. *OsMADS17* had a significant *cis*-eQTL, but similar to *SDT*, the local variants of *OsMADS17* were only weakly associated with SPP (*p* value = $7.36 \times 10^{-5}$, Fig. 7c).

The associated *cis*-variants of *OsMADS17* and *SDT* had only weak associations with SPP, but both *cis*-ECs were more significantly associated with SPP (rank = 64 and 1 for *cis*-ECs of all genes, respectively), probably because their expressions were both regulated by more than one independent *cis*-variant. The fitted *cis*-EC combined the effects of multiple *cis*-variants and thus had a higher power to detect associations compared to the associations between individual variants and phenotypes. These results demonstrate the benefit of using population-level transcriptome data to identify causal genes.

The examples of *SDT* and *OsMADS17* prompted us to ask whether the *cis*- and *trans*-ECAS genes tend to have stronger GWAS signals for panicle traits. We found that indeed the GWAS signals of *cis*-eQTLs for the *cis*- and *trans*-ECAS genes had a great departure from the null distribution compared to the GWAS signals of all variants (Fig. 6g and Supplementary Fig. 14a, b). Although the *cis*-eQTLs of the *cis*- and *trans*-ECAS gene had a relatively weak GWAS signals, the departure from the null distribution is significantly higher than (Kolmogorov-Smirnov test, *p* value = $2.8 \times 10^{-8}$) the GWAS signals of the *cis*-eQTLs of the TWAS significant gene. This result suggests that the *cis*-regulatory variants of the *cis*- and *trans*-ECAS genes have a stronger GWAS signal compared to the TWAS significant genes and *cis*- and *trans*-ECAS genes are more likely to be causal genes affecting panicle traits.

## The number of favorable alleles on *cis*-regulatory variants of *cis*- and *trans*-ECAS genes is a good predictor of panicle traits

Since the GWAS signal of *cis*-regulatory variants on phenotype is relatively weak, we wanted to know whether these *cis*-regulatory variants of *cis*- and *trans*-ECAS genes could explain the variation in phenotypes. We first defined alleles of *cis*-eQTLs that up-regulate PAGs or down-regulate NAGs as "favorable alleles" and then counted the number of favorable alleles in each variety to predict phenotypes in the remaining 254 of the 529 varieties for which the transcriptome data were not available. The results show that the number of favorable alleles of the *cis* and *trans*-ECAS genes in each variety for all three traits are highly correlated with the corresponding phenotypes (Fig. 6h and Supplementary Fig. 14c, d). These correlations are significantly higher (*p* value <0.05) than the correlations between the phenotypes and the number of favorable alleles for the same number of randomly selected TWAS significant genes with significant *cis*-genetic variance (Fig. 6i and Supplementary Fig. 14e, f). These results indicate that the *cis*-eQTL of *cis*- and *trans*-ECAS genes have better predictive power for phenotypes and also suggest that *cis*- and *trans*-ECAS genes are more likely to be causal genes affecting phenotypes in varieties. Thus, *cis*-regulatory variants of *cis*- and *trans*-ECAS genes can be used as markers for

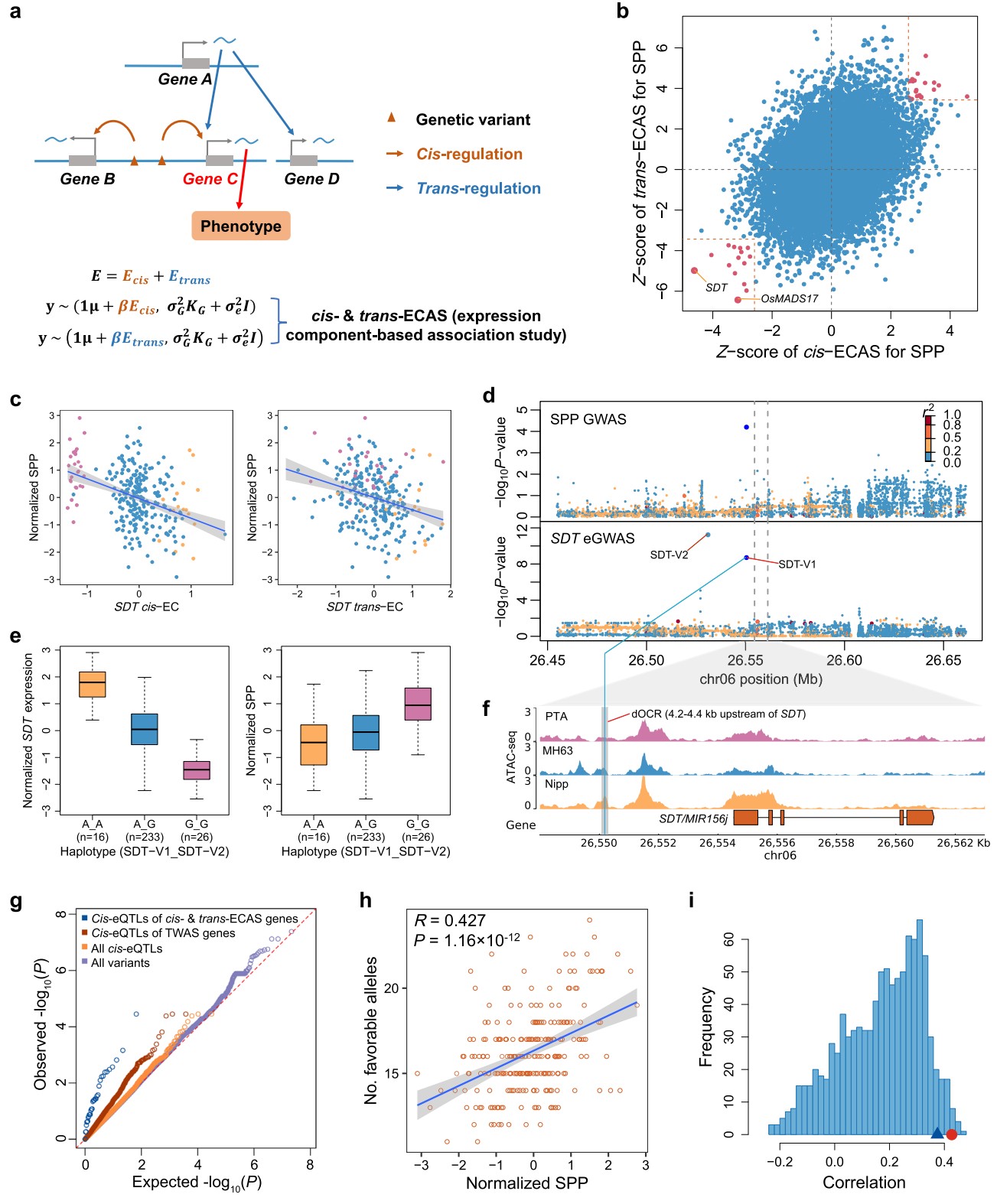

molecular breeding, and aggregation of favorable alleles at these loci may further improve rice yield.

## Cis- and trans-EC-based associations help construct regulatory networks for gene expression in panicle development

Similar to the identification of candidate genes affecting phenotypes, it is still a challenge to identify regulatory genes affecting gene expression. In the above sections, we demonstrated that *cis-*

and *trans*-ECAS can well identify genes affecting phenotypes. Similarly, we can take the expression of a target gene as a molecular phenotype (e-trait) and associate it with all genes' *cis-* and *trans*-EC to identify putative upstream regulatory genes of the target gene. Alternatively, we can also examine the association between the *cis-* and *trans*-EC of a regulatory gene with the expression of all other genes to identify putative downstream genes. In this way, we can construct causal gene regulatory networks, which utilize information

**Fig. 6 | Identification of putative causal genes by *cis*- and *trans*-expression component-based association study (*cis*- and *trans*-ECAS). a** A schematic diagram of identifying putative causal genes by *cis*- and *trans*-ECAS. See main text for details. **b** Scatter plot of the association of SPP with *cis*-EC (*x*-axis) and *trans*-EC (*y*-axis) of genes. Red dots indicate genes whose both *cis*- and *trans*-EC are significantly associated with SPP. **c** Correlations of SPP with *cis*-EC (left) and *trans*-EC (right) of *SDT*. The colors of the dots represent different haplotypes which are the same as in (**e**). The error bands indicate 95% confidence intervals. **d** Regional association plots of GWAS for SPP (top) and *SDT* expression (eGWAS, bottom) in a 200-kb region centered on *SDT*. The colors of the dots represent the linkage disequilibrium (measured by $r^2$) of each variant with the lead variant of GWAS for SPP (dark blue). *p* values of SDT-V1 and SDT-V2 were calculated using a multivariate linear mixed model with SDT-V1 and SDT-V2 as independent variables, while *p* values for the other variants were calculated using a linear mixed model with SDT-V1 and SDT-V2 as covariates[71]. **e** Box plots of *SDT* expression and SPP of different haplotypes. The *x*-axis indicates the haplotypes formed by combining genotypes of SDT-V1 (A/G) and SDT-V2 (A/G). The definitions of the box plots are the same as Fig. 4b. **f** Chromatin accessibility profiles of three varieties with different haplotypes around the *SDT*. PTA, MH63, and Nipp indicate the varieties of Padi Tarab

Arab, Minghui63, and Nipponbare, respectively, which belong to the three haplotypes in (**e**) (G_G, A_G, and A_A, respectively). The *y*-axis values are Counts Per Million mapped reads (CPM) of ATAC-seq. The gray rectangular area indicates the distal open chromatin region (dOCR) surrounding SDT-V1 (blue line), where changes in chromatin accessibility are consistent with variations in *SDT* expression. **g** Q-Q plot of GWAS signals for SPP for different groups of variants. **h** The correlation between SPP and the number of favorable alleles at *cis*-eQTLs for the *cis*- and *trans*-ECAS significant genes of SPP in each variety. The analysis included 254 varieties, for which transcriptome data were not acquired. Alleles that up-regulate PAGs or down-regulate NAGs are defined as favorable alleles. Pearson's correlation coefficient is used for the test. **i** Histogram of the correlations between SPP and the number of favorable alleles at *cis*-eQTLs for randomly selected TWAS significant genes. The same number of genes as the *cis*- and *trans*-ECAS genes were selected each time from TWAS significant genes with significant *cis*-genetic variance and repeated 1000 times. The blue triangle indicates the 0.95 quantiles of the correlations, and the red dots indicate the correlation between phenotypes and the numbers of favorable alleles at *cis*-eQTLs for the *cis*- and *trans*-ECAS genes. Source data underlying (**b**, **c**, **e**, **h**) are provided as a Source Data file.

from genetic variation and are therefore more reliable than co-expression networks.

Taking *OsMADS17* as an example. ECAS analysis indicated that it may be a causal gene that negatively regulates SPP (Fig. 7a), but its regulatory mechanism is unclear. By performing ECAS using the *cis*- and *trans*-EC of *OsMADS17* with the expression of TWAS significant genes (Methods), we found that *OsMADS17* might be an upstream regulator of 64 TWAS significant genes (Supplementary Data 13). Among them, we noticed that the *cis*- and *trans*-EC of *OsMADS17* were positively associated with the expression of *SDT* (*p* value = $3.08 \times 10^{-5}$ for *cis*-ECAS and $8.43 \times 10^{-16}$ for *trans*-ECAS, Fig. 7b). And when associating the *cis*- and *trans*-EC of all genes (as independent variables) with the expression of *SDT* (as dependent variable), the *cis*- and *trans*-EC of *OsMADS17* ranked 26th and 1st, respectively, in the ECAS results. These results suggest that *OsMADS17* might be a regulator of *SDT* and negatively regulate SPP by up-regulating *SDT* expression.

To gain more insight into the transcriptional regulation of *SDT*, we collected young panicles of three varieties with different haplotypes on *SDT* to perform the assay for transposase-accessible chromatin using sequencing (ATAC-seq). From the ATAC-seq results, we noticed that the chromatin accessibility of a distal open chromatin region (dOCR) 4.2–4.4 kb upstream of the TSS of *SDT* (around the putative causal variant SDT-V1) was consistent with expression variation of *SDT* for the three varieties (Fig. 6f). Thus we speculated that this dOCR is an important *cis*-regulatory region for *SDT*. We conducted transient expression assays using a dual-luciferase reporter system in rice protoplasts. The relative luciferase activity results showed that *OsMADS17* failed to activate the expression of the reporter gene *firefly luciferase* driven by the sequence of 0–4 kb or the dOCR upstream of the *SDT* (Fig. 7d, e). In contrast, the relative luciferase activity showed higher activity when driven by the 0–4.6 kb sequence upstream of *SDT*, suggesting that *OsMADS17* may upregulate the expression of *SDT* by regulating the dOCR upstream of *SDT*. Electrophoretic Mobility Shift Assays (EMSA) further confirmed that *OsMADS17* can directly bind to this dOCR (Supplementary Fig. 16a). In addition, qRT-PCR analysis showed that *SDT* expression was moderately down-regulated in the young panicle of the *CR-osmads17* mutant compared to the wild-type (Supplementary Fig. 16b), further demonstrating that *SDT* transcription is positively regulated by *OsMADS17*.

The expression genome-wide association study (eGWAS) results of *SDT* showed an obvious peak in the *OsMADS17* region, which also supports the regulation of *SDT* by *OsMADS17*. However, the association did not reach the genome-wide significance threshold (Fig. 7c), making it difficult to detect such regulatory relationships by traditional eQTL analysis. In contrast, *cis*- and *trans*-ECAS can efficiently identify the

regulatory relationship between *SDT* and *OsMADS17*, confirming the efficacy of our approach in identifying gene regulators.

Besides being found to regulate *SDT*, *OsMADS17* may also negatively regulate *RFL* and *OSH1* (Fig. 7j and Supplementary Data 13), but whether the regulatory effect of *OsMADS17* on these genes is direct or indirect requires further investigation.

## Validation and breeding application of *OsMADS17*

As *OsMADS17* has not been reported to impact SPP in rice, we validated its function using the CRISPR/Cas9 system. We observed a 19.2% increase in SPP and a 25.2% increase in the number of secondary branches (NSB) compared the knock-out line *CR-osmads17* to wild type, while no significant difference was found in NPB (Fig. 7f–i).

We further analyzed the natural variation of *OsMADS17*. We found two variants in the coding region of *OsMADS17*, one of which is a synonymous SNP (vg0429308534) and the other (vg0429308367) is a 65-bp long sequence inserted at the start codon of the ORF, potentially shifting the ORF backwards by 15 bp. However, these two variants are not associated with the SPP (*p* value >0.05). We also identified 12 variants within 5 kb upstream of the TSS of *OsMADS17* that were significantly associated with its expression level (*p* value <$10^{-5}$). Based on all 14 variants, we identified three main haplotypes in the *OsMADS17* region (Supplementary Fig. 15c, d), with significant differences in *OsMADS17* expression and SPP between haplotypes. Among the variants of *OsMADS17*, a multi-allelic variant with 11-bp and 39-bp deletions (named OsMADS17-V1, composed by vg0429307485 and vg0429307502) coincides to the three haplotypes and is most significantly associated with the expression level of *OsMADS17* (*p* value = $5.78 \times 10^{-18}$). This variant also has the highest functional impact score among the 12 non-coding variants predicted by a deep learning-based approach in RiceVarMap[46], suggesting that it may be a functional variant that leads to changes in *OsMADS17* expression. At this site, we observed that both types of deletions were associated with down-regulation of *OsMADS17* and an increase in SPP (Fig. 7a and Supplementary Fig. 15d), indicating that they may be superior alleles for breeding.

To explore whether the superior alleles of *OsMADS17* have breeding value, we introgressed the haplotype 3 of *OsMADS17* (carrying a 39-bp deletion on OsMADS17-V1 and with the highest SPP) derived from the tropical *japonica* variety IRAT109 into a breeding line 1035 with temperate *japonica* background, which carries the reference genotype of *OsMADS17*. We selected a line heterozygous on OsMADS17-V1 in a $BC_4F_3$ population and examined the phenotypes of its segregating progenies based on the genotypes of OsMADS17-V1. The results showed a 25.0% increase in NSB and an 18.5% increase in

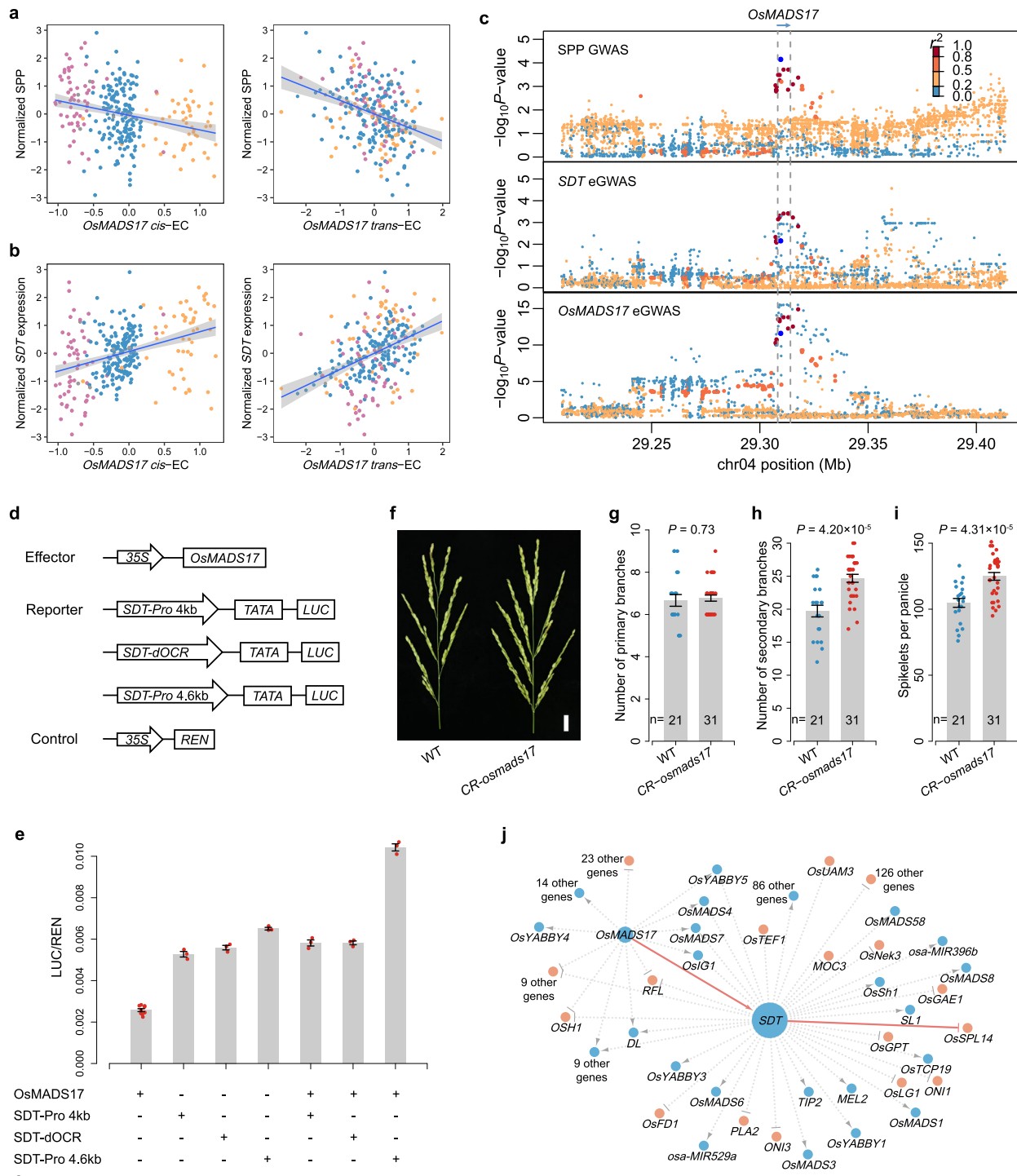

**Fig. 7 | *OsMADS17* regulates *SDT* transcription and affects SPP. a** Correlations of SPP with *cis*-EC (left) and *trans*-EC (right) of *OsMADS17*. The colors of the dots represent different genotypes of the OsMADS17-V1 variant: reference type (orange), 11-bp deletion type (blue), and 39-bp deletion type (purple). The error bands indicate 95% confidence intervals. **b** Correlations of *SDT* expression with *cis*-EC (left) and *trans*-EC (right) of *OsMADS17*. The colors of the dots represent different genotypes of OsMADS17-V1, as in (**a**). **c** Regional association plots of GWAS using linear mixed model for SPP (top), *SDT* expression (middle) and *OsMADS17* expression (bottom) in a 200-kb region centered on *OsMADS17*. **d** Schematic diagram of the effector and reporter constructs used for transient transcriptional activity assay in (**e**). Firefly luciferase gene *LUC*, driven by the 4-kb and 4.6-kb promoters of *SDT* as well as dOCR (4.2−4.4 kb upstream of *SDT*, as shown in Fig. 6f), was used as the reporter. **e** Transient transcriptional activity assay showing that OsMADS17 transactivates *SDT* transcription and this transactivation requires the dOCR upstream of *SDT*. All data are means ± SEM (9 biologically independent samples for OsMADS17 + control and 3 for the others). **f** Panicle morphologies of *CR-osmads17* mutant and WT. WT wild-type. Scale bar, 2 cm. **g**−**i** Quantification of the number of primary branches (**g**), number of secondary branches (**h**), and spikelets per panicle (**i**) in the main panicle of *CR-osmads17* mutant and WT. Data are means ± SEM. Comparisons are made by two-tailed Student's *t* test. **j** Gene regulatory networks constructed using *cis*- and *trans*-ECAS. The SPP TWAS-significant genes (FDR <0.01) were used as e-traits, and the network centered on *SDT* and *OsMADS17* (*SDT* and *OsMADS17* as regulatory genes) is shown. The color of the nodes indicates the regulatory direction of genes for SPP: light salmon for positive correlation and blue for negative correlation. The size of each node circle indicates the number of nodes it connects to. Solid lines represent the regulatory relationships validated in this study and those previously reported. Source data underlying (**a**, **b**, **e**, **g**−**i**) are provided as a Source Data file.

SPP for lines with the 39-bp deletion on OsMADS17-V1 compared to lines with the reference genotype (Supplementary Fig. 16c–f), suggesting that this superior allele of *OsMADS17* could be used to improve NSB and SPP in rice varieties.

We examined the distribution of the 39-bp-deletion allele of *OsMADS17* in varieties using RiceVarMap[46]. This superior allele has the highest frequencies in tropical *japonica* (0.905) and lower frequencies in temperate *japonica* and *indica* (0.087 and 0.028, respectively) which have larger planting areas. Similarly, the superior haplotype of *SDT* (G_G) has a higher frequency in tropical *japonica* (0.687) but a lower frequency in other subgroups (<0.1). These results suggest that the superior alleles of *OsMADS17* and *SDT* have not been widely used in breeding and have great potential for enhancing grain number in the main cultivars.

## Discussion

In this study, we identified a large number of genes whose expression levels were associated with panicle traits in rice for the first time. Although association is not causation, these information provide new insights and resources to further resolve the regulatory networks of panicle development. Among those genes associated with SPP, NAGs are enriched in negative regulators of SPP and might provide abundant targets for improving rice yield using genome editing techniques such as CRISPR-Cas[47]. Meanwhile, which of these associated genes are causative and the regulatory networks among these genes are unclear, appropriate methods are needed to overcome these limitations.

In human medical studies, several methods have been developed in recent years to identify causal genes affecting phenotype based on the correlation between the *cis*-component of gene expression and the phenotype[48]. However, genes identified by this approach are also prone to be false positives due to LD between *cis*-regulatory variants and phenotypic causal variants[49,50]. We developed a method to integrate the association of *cis*- and *trans*-components of gene expression with phenotype to screen for causal genes affecting phenotypes or gene expression. This method is particularly useful for crop research, as it is relatively easy to obtain expression data and phenotype data for the same population simultaneously. To enhance the power of identifying causal genes, we recommend: (1) integrating phenotype data across years or locations; (2) obtaining phenotype data from samples grown in the same environment as transcriptome samples; (3) acquiring high-quality transcriptomes from as many varieties as possible.

We found that *cis*-variants of *SDT* and *OsMADS17* affect panicle traits and further demonstrated that *OsMADS17* regulated the expression of *SDT*. Since *SDT* encodes *MIR156j* which is the negative regulator of *IPA1*[5,6], our study reveals a regulatory mechanism by which *OsMADS17* up-regulates *SDT* and thus negatively regulates *IPA1* to reduce SPP. The superior alleles of *OsMADS17* and *SDT* might provide a useful resource for breeding. In addition, despite the weak GWAS signals of *SDT* and *OsMADS17* (*p* value >10⁻⁵), the results of *cis*- and *trans*-EC associations strongly support that they are causal genes, demonstrating the ability of our method to identify causal genes in small-effect loci. The identified *cis*- and *trans*-ECAS genes are worth further experimental validation.

The genomic footprint of domestication or breeding processes in crops has been widely studied[39,51,52], but whether *cis*-regulatory variants affecting gene expression are under selection during domestication or breeding, and how the selection affects agronomic traits in crops has rarely been studied. We found that the variants with higher DAF are more likely to be *cis*-eQTLs of genes, indicating that *cis*-regulatory variants of gene expression were subject to positive selection, which is consistent with our previous study finding that high-impact regulatory variants in rice are subject to positive selection[46]. However, unlike the study in maize finding that domestication and breeding processes

prefer to increase gene expression[41], we found that the selection direction of the derived alleles of *cis*-eQTL is differentiated and related to gene function. The derived alleles with high-frequency prefer to up-regulate the expression of PAGs or down-regulate the expression of NAGs, i.e., tend to have a positive effect on SPP. In contrast, derived alleles of *cis*-eQTL which have a negative effect on SPP tend to have lower frequencies. This difference in selection is further supported by the difference in GWAS signals for lead variants of *cis*-eQTL with positive and negative effects of the derived alleles on the SPP phenotype. This indicates that domestication and improvement rewired gene expression to increase grain numbers in rice.

In conclusion, our study provides insights into the complex genetic structure and regulatory networks of panicle architecture and has potential implications for molecular breeding in rice. In addition, the methods and strategies developed in our study can efficiently identify causal genes affecting agronomic traits, even those at small effect GWAS loci, which are expected to facilitate research in other crops as well.

## Methods

### Plant materials, RNA extraction and sequencing

A set of 529 varieties was used in our previous study[18,53]. The varieties were selected to represent both the genetic diversity in Asian cultivated rice (*O. sativa* L.) and their usefulness in breeding. Based on genetic distance among varieties, we further selected 300 varieties from them to obtain young panicles. We first selected a fixed set of varieties (those with high quality genomes) and then iteratively selected the variety with the maximum distance to any variety in the current set of varieties. This process continues until 300 varieties have been selected. These varieties were planted in four batches in the experimental field of Huazhong Agricultural University during the summer at Wuhan (30.4°N, 114.2°E), China. One variety, Zhenshan 97, was planted in all four batches and its young panicles were collected each time. Total RNA was extracted from young panicles (1–2 mm) of 275 rice varieties using the TRIzol reagent (Invitrogen). Totally 278 paired-end libraries (one library for each of 274 varieties and four libraries for Zhenshan 97) were constructed and sequenced using the Illumina HiSeqX platform at Novogene to obtain >9 Gb of PE150 sequencing data for each library. The high correlations between the four replicates of Zhenshan 97 indicate that the effect of planting batch on the transcriptome is small (Supplementary Fig. 5a–d).

### RNA-seq data processing

We first counted the read numbers of RGAP (Rice Genome Annotation Project) annotated genes[54] for each sample separately using Salmon[55]. To quantify miRNAs, we extended the primary transcript sequences of miRNAs annotated in miRBase[56] by 200-bp each to the left and right as reference transcripts and used Salmon to quantify the read numbers of the extended miRNA transcripts. Then the read numbers of genes and miRNAs were incorporated into one file for each sample, and the TPMs (transcript per million) were calculated to correct for differences in library size across samples. The TPMs of different transcripts belonging to the same gene were combined as the gene's TPM. Then the TPMs were added by one, and log2 transformed as gene expression levels. We screened genes with expression levels greater than 0.5 in more than 14 varieties (5% of the 275 varieties) as expressed genes and obtained 30,869 expressed genes. To control system bias, we used PEER[57] to exclude the first three factors, and the residuals were used as corrected gene expression levels. Then a rank-based approach was used to transform the distribution of expression levels of each gene in the population into a normal distribution.

### Phenotype data processing

The 275 rice varieties for which young panicle transcriptome data were obtained were also investigated for panicle phenotypes. Phenotypic

data for each variety were obtained from five independent plants. The main panicles of five independent plants were collected, the primary and secondary branches were separated with tape and pasted on white paper, and RGB pictures were taken and saved by a camera. Further statistics were conducted on spikelets per panicle (SPP), number of primary branches (NPB), and panicle length (PL). To integrate published phenotypic data from Hainan 2013, Wuhan 2013, and Wuhan 2014[12], We used the lmer function in the R package lme4[58] to fit the best linear unbiased prediction model (BLUP) with both year and variety as random effects, and then used a rank-based approach to transform the phenotypic values of the BLUP to a normal distribution.

### Genome-wide association studies for panicle traits

Genotype data were obtained from RiceVarMap V2[46], and variants with minor allele frequency (MAF) greater than 0.05 and the number of minor alleles greater than six were used for subsequent analysis, with a total of 10,567,425 variants in the 529 varieties panel. We used linear mixed models for association studies, EMMAX[59] was used to implement the calculations, and all variants with MAF greater than 0.05 were used to calculate the kinship matrix by PLINK[60]. The genome-wide significance threshold for GWAS was determined by the significance threshold corrected for multiple hypothesis testing, and the GEC[61] showed a total of 1,970,782 independent variants in 529 varieties panel, and the number of independent variants in 275 varieties panel for which transcriptome data were obtained was quite similar (1,967,882), thus the genome-wide significance threshold was set at $2.54 \times 10^{-8}$ and the suggestive threshold was set at $5.07 \times 10^{-7}$.

### Transcriptome-wide association studies for panicle traits

The panicle traits were associated with the expression levels of each expressed gene to uncover panicle development-related genes. The following linear mixed model were used:

$$y \sim (\beta x, \sigma_G^2 K_G + \sigma_\epsilon^2 I) \qquad (1)$$

Here, $y$ denotes the phenotypes of a trait across varieties, $x$ denotes the expression levels of a gene across varieties, $\beta$ is the effect of the gene on the phenotype, and $K_G$ is the kinship matrix. EMMAX[59] was used to implement the calculations. Genes with FDR less than 0.05 were considered to be significantly associated with the trait.

### Analysis of variance compositions of panicle traits and TWAS significant genes

SNP heritability for panicle traits was estimated using GCTA[20]. All common variants were used to calculate the genetic relationship matrix (GRM), with the parameter "--autosome --maf 0.05 --make-grm-inbred". Then, the restricted maximum likelihood method was used to estimate SNP heritability. Genetic correlations between the expression of genes and panicle traits were also estimated using GCTA, with the parameter "--reml-bivar 1 2 --reml-bivar-nocove --reml-bivar-lrt-rg 0". To estimate the phenotypic variances explained by GWAS loci identified at different thresholds, we first screened all lead variants which association $p$ values smaller than $10^{-3}$. Then regression models for phenotypes were constructed using variants exceeding different $p$ value thresholds ($10^{-7}$, $10^{-6}, 10^{-5}, 10^{-4}, 10^{-3}$), respectively. The regression models were fitted by LASSO and implemented using the cv.glmnet function in the glmnet package[62], and the 10-fold cross-validation $R^2$ was employed to represent the explained phenotypic variance. To evaluate the significance of phenotypic variances explained by the GWAS loci identified at different thresholds, we permuted the phenotypes of panicle traits 500 times and performed GWAS for each permuted trait. Then, the same method was used to estimate the variance of the permuted phenotypes explained by the GWAS loci identified at different thresholds. To demonstrate that the expression variations of TWAS significant genes affecting the panicle traits are mainly contributed by small-effect GWAS loci, we

predicted the expression of the top 500 TWAS significant genes in different varieties using the GWAS loci identified at different thresholds ($10^{-7}, 10^{-6}, 10^{-5}, 10^{-4}, 10^{-3}$). The regression model was fitted by LASSO for each gene at each GWAS threshold. Then, the $R^2$ (square of Pearson's correlation coefficient between predicted gene expression and panicle trait) was used to represent the genetic-caused expression variation of TWAS significant gene affecting panicle trait. For the permutation test, TWAS was also performed for each of the permuted trait, and then the same analysis as for the panicle trait was performed using the top 500 TWAS genes and the GWAS loci of the permuted trait.

### Enrichment analysis of TWAS significant genes

Gene Set Enrichment Analysis (GSEA) was performed using the GSEApy package[36]. We calculated the ratio of the average expression in young panicles to the average expression in other tissues for each gene, genes with a ratio greater than 0.95 quantiles of all genes were considered to be preferentially expressed in young panicles. The genes preferentially expressed in young panicles in ZS97 and MH63[43] and Nipponbare[63] were merged. Gene sets with progressively up-regulated and progressively down-regulated expression in young panicles were from the CREP[43]. The list of transcription factors is from PlantTFDB[64], and the list of chromatin modification-related genes is from Ensembl BioMarts[65]. Fisher's exact test was used to examine the TWAS significant genes enriched transcription factor family. GO enrichment analysis and TF binding motif enrichment analysis of TWAS significant genes were performed on the web service of Plant Regulomics[37].

### pQTL-eQTL hotspots of TWAS significant genes

To identify pQTL-eQTL hotspots that affect phenotype and regulate multiple TWAS significant genes, we performed an association analysis of GWAS pQTLs (GWAS $p$ value $<10^{-4}$) with the expression level of all expressed genes. We then analyzed the enrichment of associated genes ($p$ value $<10^{-4}$ for the association between pQTL and gene expression) with TWAS significant genes for each pQTL. The pQTLs whose associated genes were significantly enriched for TWAS significant genes were defined as pQTL-eQTL hotspots (number of overlapped genes >10, BH-adjusted Fisher's exact test $p$ value <0.05).

### Derived allele frequencies of lead variants of *cis*-eQTLs

To assess the effect of rice domestication and breeding improvement processes on gene expression, we obtained the derived alleles and allele frequencies of variants from RiceVarMap V2[46]. Alleles that differ from the major allele of the wild rice population[42] were defined as derived alleles. Variants with missing rates greater than 0.5 in wild rice or variants in which both the major allele and minor allele in 529 germplasm were not identical to the ancestral allele (major allele in wild rice) were removed. For the remaining variants, a total of 1,275,135 variants located within 10 kb of the TSS of genes were used for *cis*-eQTL analysis. The most significantly associated variant for each gene was taken as the lead variant of *cis*-eQTL, and a total of 13,877 genes were identified with *cis*-eQTL under the threshold of $10^{-5}$.

### *Cis*- and *trans*-expression component-based association study (*cis*- and *trans*-ECAS) for panicle traits

We propose a novel strategy to identify causal genes that affect phenotype. We first calculated the genetic relatedness matrix (GRM) using variants within 100 kb of the gene. Then, based on the GRM, BLUP was performed for the expression levels of each gene using GCTA[20] to obtain the gene expression levels predicted by *cis*-genetic variations (defined as *cis*-expression component, abbreviated as *cis*-EC). We did not estimate *trans*-expression component (*trans*-EC, gene expression levels predicted by *trans*-genetic variations) directly, but used the residuals of gene expression after subtracting *cis*-EC as *trans*-EC, since estimating *trans*-EC of genes may be imprecise[66]. And the plant materials for our transcriptome data of all varieties were planted in the

same field, thus the variation in gene expression after removal of *cis*-EC was mainly from *trans*-EC and random errors. At the threshold of $1.62 \times 10^{-6}$ (Bonferroni corrected *p* value <0.05), 14,392 genes had significant *cis*-genetic variance and were used for subsequent analyses. We then used the following linear mixed models to associate the panicle traits with the *cis*- and *trans*-EC of each gene, respectively:

$$y \sim (\beta E_{cis}, \sigma_G^2 K_G + \sigma_e^2 I), \qquad (2)$$

$$y \sim (\beta E_{trans}, \sigma_G^2 K_G + \sigma_e^2 I) \qquad (3)$$

Here, *y* denotes the phenotypes of a trait across varieties, $E_{cis}$ denotes the *cis*-EC of a gene across varieties, $E_{trans}$ denotes the *trans*-EC of a gene across varieties, and $K_G$ is the kinship matrix. Then the genes for which both *cis*-EC and *trans*-EC were associated with the phenotype were prioritized as candidate genes (*cis*-ECAS *p* value <0.01, *trans*-ECAS FDR <0.01).

### Identification of upstream regulators of genes by *cis*- and *trans*-ECAS using gene expression as phenotypes

We applied the same method as for identifying causal genes for panicle traits, *cis*- and *trans*-ECAS, to identify upstream regulators of genes. The expression values of each TWAS significant gene (FDR <0.01) were used as molecular phenotype (e-trait). We associated the e-trait with the *cis*- and *trans*-EC of all other genes with significant *cis*-genetic variance, and the genes for which both *cis*- and *trans*-EC were significantly associated with e-trait (*cis*-ECAS *p* value <0.001, *trans*-ECAS FDR <0.001) were prioritized as possible upstream regulators.

### Dual-luciferase reporter assay

The coding sequences of *OsMADS17* were amplified with primer pair NONE-OsMADS17-F/R and then cloned into NONE to generate NONE-*OsMADS17* as the effector. The *SDT* truncated promoters were amplified with primer pair SDT-promoter-4kb-F/R, SDT-promoter-4.6kb-F/R, and SDT-dOCR-F/R, and then cloned into 190-LUC, respectively, to generate 190-LUC-*SDT-Pro* 4kb, 190-LUC-*SDT-Pro* 4.6kb, 190-LUC-*SDT-dOCR*, as reporter. The sequences of primers are listed in Supplementary Data 14. For each assay, the plasmids containing reporters and effectors were co-transformed with 35S-REN by the ratio of 6:6:1 in ZH11 green seedling protoplasts, with REN activity as the internal control. After incubating for 12–16 h at 25 °C, the relative luciferase activity was measured using the DLR assay system (Promega) and the TECAN Infinite M200 microplate reader.

### Creation of *osmads17* mutant

To obtain the *osmads17* mutant, the sgRNA-CAS9 plant expression vectors were constructed as described previously[67]. Two target sequences (Supplementary Data 14) of the guide RNA were designed to target the *OsMADS17* gene. Above the constructs were introduced into *Agrobacterium tumefaciens EHA105* and were transformed into the callus derived from japonica cultivar ZH11 by Agrobacterium-mediated transformation as previously described[68]. Primers (Supplementary Data 14) were designed to amplify target DNA fragments of transgenic plants and PCR products were sequenced using Sanger sequencing to determine genotypes.

### EMSA

For EMSA, two complementary oligonucleotides were synthesized, labeled with biotin at 5' end, and annealed to form DNA probes. The corresponding unlabeled DNA probes were used as competitors. EMSA was performed using the LightShift Chemiluminescent EMSA Kit (Thermo Fisher Scientific). In the experimental group, 20 fM biotin-labeled probes were incubated with HIS-OsMADS17 in the binding buffer (10 mM Tris, 50 mM KCl, 1 mM EDTA, 5 mM MgCl$_2$, 1 mM DTT,

50 ng/μL Poly [dI.dC], 2.5% Glycerol, and 0.05% NP-40) for 30 min at room temperature, meanwhile, HIS protein was used as a negative control. In competition reaction, 2 pM (100×), 4 pM (200×) and 8 pM (400×) un-labeled probes were mixed with 20 fM biotin-labeled probes, the mixture incubated with HIS-OsMADS17 protein in the binding buffer for 30 min at room temperature. The DNA-protein complex was subjected to 6% native polyacrylamide gel electrophoresis at 4 °C. After gel electrophoresis separation, the biotin-labeled probes were detected using the Chemiluminescent Nuleic Acid Detection Module (Thermo Fisher Scientific) according to the manufacturer's protocol. All the primers for EMSA were listed in Supplementary Data 14.

### Quantitative RT-PCR

Total RNA was extracted from young panicle (3 mm) of rice using the TRIzol reagent (Invitrogen) according to the manufacturer's instructions, and first-strand cDNA was synthesized from 1–5 μg total RNA with HiScript III 1st Strand cDNA Synthesis Kit (+gDNA wiper) (Vazyme, R312-01). qRT-PCR was performed with the Applied Biosystems 7500 real-time PCR detection system using SYBR Green MasterMix (Applied Biosystems). The data were analyzed using the relative quantification method[69]. *Ubiquitin* was used as a control for normalization. All the primers used for qRT-PCR were listed in Supplementary Data 14.

### Reporting summary

Further information on research design is available in the Nature Portfolio Reporting Summary linked to this article.

## Data availability

Data supporting the findings of this work are available within the paper and its Supplementary Information files. RNA-seq data generated in this study have been deposited in the Genome Sequence Archive under the BioProject accession number PRJCA012684. Genotype data, variant impact scores, and subgroup classification of the varieties are available in RiceVarMap V2 (https://ricevarmap.ncpgr.cn/). Rice genome sequence and gene annotation information were obtained from RGAP (http://rice.uga.edu/). Annotation information for microRNAs was obtained from miRbase (https://mirbase.org/). The expression profiles of the entire life cycle are available in CREP (http://crep.ncpgr.cn/) for ZS97 and MH63, and in RiceXPro (https://ricexpro.dna.affrc.go.jp/) or Gene Expression Omnibus (GSE21494, GSE39426, GSE39427, GSE39432) for Nipponbare. The list of transcription factors is available in PlantTFDB (http://planttfdb.gao-lab.org/). The list of chromatin modification-related genes is available in Ensembl BioMarts (http://plants.ensembl.org/biomart/martview/). The genotype data of wild rice germplasm used to identify derived alleles are available at RiceHap3 (http://server.ncgr.ac.cn/RiceHap3/), and the sequencing data are available at European Nucleotide Archive under accession number ERP001143 (https://www.ebi.ac.uk/ena/browser/view/PRJEB2829). Source data are provided with this paper.

## Code availability

The codes for identifying causal genes affecting phenotypes and for constructing regulatory networks of functional genes have been deposited in Zenodo (https://doi.org/10.5281/zenodo.10004834) and in Github (https://github.com/Minglc/CisTrans-ECAS)[70].

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

## Acknowledgements

We thank Professor Liang Guo at Huazhong Agricultural University for his helpful comments on the manuscript. The computations in this paper were run on the supercomputing system in the Supercomputing Center of Wuhan University and bioinformatics computing platform of the National Key Laboratory of Crop Genetic Improvement, Huazhong Agricultural University. This work was supported by grants from the National Natural Science Foundation of China (31922065 to W.X.; 31821005, 32261143466, 31630054 to C.W.), the Natural Science Foundation of Hubei Province (2023AFA043 to W.X., 2022CFA024 to C.W.), the National Key Research and Development Program of Hubei Province (2022BBA54 to C.W.), HZAU-AGIS Cooperation Fund (SZYJY2023003 to W.X.), and the Foundation of Hubei Hongshan Laboratory (2021hszd010 to C.W., 2021hszd005 to W.X.).

## Author contributions

W.X. and C.W. designed and supervised this study. D.F., T.X., X. Xiong, M.L., Y.Z., G.L., X. Xu, C.X., R.Z., K.L., Z.W., J.M., L.Y., Q.Y., W.C., and X.L. performed experiments. L.M., H.Z., J.W., S.L., Y.L., and X.W. analyzed data. L.W. helped supervise the study. L.M., D.F., C.W., and W.X. wrote the manuscript with input from all authors. All the authors read and approved the manuscript.

## Competing interests

The authors declare no competing interests.
