## [Peer Review File · Nature Communications]

Transcriptome-wide association analyses reveal the impact of regulatory variants on rice panicle architecture and causal gene regulatory networksREVIEWER COMMENTS

Reviewer #2 (Remarks to the Author):

It is a challenge to identify the causal genes and to construct regulatory networks for the traits regulated by multiple small-effect genes in crops. Here, the authors used transcriptome-wide association analyses to reveal the impact of regulatory variants on rice panicle architecture, and developed a method to identify causal genes at even small-effect GWAS loci and to construct regulatory networks controlling panicle architecture.

1. Transcriptome of the 1-2 mm young panicles from 275 rice varieties were used in this study, the data can effectively explore the causal genes. The more varieties used will improve the accuracy of results, but it will increase the costs. Did have any test the minimal rice varieties used that can effectively explore the causal genes? It might be better to discuss it.

2. The authors showed that panicle traits were highly consistent across locations and years, suggesting the related gene expression might be influenced limitedly by environmental factors. How about the effectiveness of the here developed method to detect the causal genes for traits that influenced by environmental factors?

3. The authors observed that the transcription factors were mainly involved in the panicle traits by transcriptome-wide association studies (TWASs). A larger number of TWAS significant genes were identified in the pQTL-eQTL hotspots, but it is still not easy to reveal the regulatory genes in these hotspots. The transcription factors might be the candidate genes, while any other novel factors could be explored using this method?

4. The authors indicated that OsMADS17 affects SPP by regulating SDT transcription. It might be better to provide evidence that OsMADS17 and SDT function in the same genetic pathway to regulate SPP if possible. What is the meaning of the nodes circle size in Figure 7J?

5. The significant difference test should be added in the Supplemental Figure 15: Haplotype analysis of SDT and OsMADS17.

Reviewer #3 (Remarks to the Author):

Grain yield of rice is a quantitative trait which is mainly determined by tiller number, grain number and grain weight. In particular, the number of grains per panicle (panicle branching) is controlled by many quantitative trait loci (QTL), but it's also largely influenced by multiple environmental factors. In the past decades, several genes that control shoot apical meristem development and panicle branches have been identified and characterized by using GWAS, QTL analysis and map-based cloning approaches, however, the mechanisms underlying the interplay among these key regulators/genes still remain unclear. In this manuscript, the authors performed transcriptome-wide association analysis (TWAS) combined with GWAS and quantitative trait loci mapping, and identified new regulators of panicle branching and grain numbers. The authors also developed a method to identify the candidate genes and their regulatory works, and identified 36 putative new genes in regulating panicle branching. This new approach will help us to improve efficiency and maximize the genetic load of factors modulating quantitative traits in crops. The topic of this manuscript is interest for researchers follow trends and understanding of the genetic control of panicle architecture. However, there are deficiencies in experimental analysis that require further substantiation before the manuscript can be acceptable for publication.

1, previous studies by using integrating GWAS, eQTL, chromatin landscapes and/or gene expression have revealed the genetic control of key agronomic traits in many crops. Although many genes have been reported to have cis- (or trans-) variants that regulated gene transcription, and thus causing variation of grain yield-related traits. To in-depth investigate the genetic basis and gene regulatory networks in the regulation of panicle architecture, the authors should combine with looking for epigenetic DNA methylation and chromatin landscapes relative to genome-wide

transcription analysis of 275 rice varieties.

2, according to the GWAS results of panicle architecture, the authors suggested that variations in panicle branching and grain number appear to be mainly regulated by many small-effect loci. In fact, unlike thousand-grain weight, panicle branching and grain number are largely affected by multiple environmental factors. I don't know if the TWAS results could be repeated from the same population which were planted in the different fields or geographic areas (or different years).

3, in Fig.7, the authors showed that OsMADS17 affected panicle architecture through regulating miR156j transcription. The authors only showed the panicle phenotype of the *osmads17* mutant, they did not present the evidence for supporting the genetic interaction between OsMADS17 and miR156j.

4, the reference of SDT (miR156j) should be cited in the text

We are very grateful to the reviewers for their careful review and valuable comments. These comments are very helpful in revising and improving our paper. We have carefully considered the comments and revised the manuscript according to the suggestions.

The major revisions in the manuscript are highlighted in red. The most significant modification is the addition of two experiments: Electrophoretic Mobility Shift Assay (EMSA) demonstrating *OsMADS17* directly binding to the upstream dOCR of *SDT*, and qRT-PCR analysis of *SDT* expression in young panicles of *Osmads17* mutants and WT. These experiments provide additional molecular and genetic evidence to support the conclusion that *OsMADS17* directly regulates the expression of *SDT*.

We also corrected the manuscript's formatting according to the *Nature Communications* guidelines, including the labeling of figure panels (changing uppercase letters to lowercase letters) and the citations. Additionally, the Supplementary Table was renamed as Supplementary Data. During the preparation of the source data, we found and corrected several errors in the figures of the manuscript: (1) swapped Figure 2d and Figure 2e, as well as Supplementary Figure 3d and Supplementary Figure 3e, due to the misplacing of the results of SPP and NPB when combining the figures; (2) redrew Figure 6g, 6h, and 6i, because the genes identified by *cis-* & *trans-*ECAS included both *SDT* and *miR156j*, which are the same loci, and the results were now recalculated after removing the redundancy. It is important to note that these corrections do not influence the conclusions of the manuscript.

We would like to acknowledge the contributions of two new authors, Qian Yu and Wenqi Chai, who provided valuable assistance in conducting the experiments. Furthermore, in response to editorial requests, we added a Data Availability statement to the manuscript and attached the source data of the figures.

The detailed point-by-point response to the reviewers' comments are described below. Thank you very much for your time and effort in reviewing our manuscript.

Reviewer #2 (Remarks to the Author):

It is a challenge to identify the causal genes and to construct regulatory networks for the traits regulated by multiple small-effect genes in crops. Here, the authors used transcriptome-wide association analyses to reveal the impact of regulatory variants on rice panicle architecture, and developed a method to identify causal genes at even small-effect GWAS loci and to construct regulatory networks controlling panicle architecture.

1. Transcriptome of the 1-2 mm young panicles from 275 rice varieties were used in this study, the data can effectively explore the causal genes. The more varieties used will improve the accuracy of results, but it will increase the costs. Did have any test the minimal rice varieties used that can effectively explore the causal genes? It might be better to discuss it.

Response: The question raised by the reviewer regarding the minimum number of rice varieties required for effective exploration of causal genes is complex. As the reviewer mentioned, "The more varieties used will improve the accuracy of results". In GWAS, the power to detect true genetic associations is influenced by many factors, such as sample size, trait heritability, population structure, effect size of genetic variants, minor allele frequency, and significance level. For identifying causal genes using TWAS, estimating power is even more challenging, as it is also influenced by factors such as the heritability of e-traits, the quality of transcriptome data, and the approach used, as described by He et al (Statistical power of transcriptome-wide association studies. *Genet Epidemiol*, 2022, 46:572-588).

We tested the capability of our method to detect causal genes at different transcriptome sample sizes by randomly selecting different numbers of rice varieties. Under the same threshold employed in this study, when the sample size is less than 200, we observed an obvious reduction in the number of genes with significant *cis*-genetic variance (genes whose expression levels can be accurately predicted by *cis*-genetic variants) and causal genes associated with SPP (Response Fig. 1a-d). However, the two causal genes, *SDT* and *OsMADS17*, can still be prioritized in 73 *japonica* varieties (Response Fig. 1e), probably because the superior haplotypes of these two genes have relatively high frequencies in tropical *japonica*. These results suggest that it is difficult

to have certain criteria for determining the minimum sample size that can effectively identify causal genes, but we recommend obtaining transcriptomes of more samples when the budget permits and the quality of the samples is assured.

We have incorporated these results into the Discussion.

Response Fig. 1: (a-d) Number of genes with significant *cis*-genetic variance (a) and SPP-associated genes via *cis*-ECAS (b), *trans*-ECAS (c), and both (d) across different numbers of randomly sampled rice varieties (n=100, 150, 200; each with 3 replicates). (e) Scatter plot of the association of SPP with *cis*-ECAS (*x*-axis) and *trans*-ECAS (*y*-axis) of genes in 73 *japonica* varieties.

2. The authors showed that panicle traits were highly consistent across locations and years, suggesting the related gene expression might be influenced limitedly by environmental factors. How about the effectiveness of the here developed method to detect the causal genes for traits that influenced by environmental factors?

Response: The method developed in our study could detect the causal genes for traits that influenced by environmental factors. In fact, as the reviewer #3 pointed out, compared to grain weight, panicle branching and grain number are largely affected by environmental factors. In our article, we stated that "these traits were highly consistent

across locations and years", but this does not mean that panicle traits and the related gene expressions are limitedly influenced by environmental factors. We simply intended to convey that our phenotype data is of high quality. We tested TWAS for panicle traits across years, and as shown in Response Figs. 2-4, TWAS results for phenotypes from different years and locations are generally consistent (Response Figs. 2-4), but for some genes, their significance and ranking varies between different years and locations.

Regarding the effectiveness of our method in detecting causal genes for traits influenced by environmental factors, we suggest that the key is to obtain high-quality transcriptome and phenotype data. On the one hand, it is important to minimize batch or systematic biases among samples and obtain transcriptome data from tissues that are crucial for determining the phenotype. This is important for helping to identify causal genes. On the other hand, integrating multi-environmental phenotype data from different years or locations can improve the robustness of association results. If multi-environmental phenotype data is not available, it is best to use phenotype data from samples grown in the same environment as those for which transcriptomes were obtained.

We have incorporated these results into the Discussion.

3. The authors observed that the transcription factors were mainly involved in the panicle traits by transcriptome-wide association studies (TWASs). A larger number of TWAS significant genes were identified in the pQTL-eQTL hotspots, but it is still not easy to reveal the regulatory genes in these hotspots. The transcription factors might be the candidate genes, while any other novel factors could be explored using this method? Response: In our study, TWAS (discovering genes whose expression are associated with traits) and *cis-* & *trans*-ECAS (identifying causal genes whose expression influence traits or e-traits) are designed to identify genes with variation in transcriptional levels, which are not necessarily transcription factors, but may also be other types of regulators such as histone modifiers, miRNAs and kinases. For example, the causal gene *SDT* we identified is a precursor gene of microRNA *miR156*.

4. The authors indicated that *OsMADS17* affects SPP by regulating *SDT* transcription. It might be better to provide evidence that *OsMADS17* and *SDT* function in the same genetic pathway to regulate SPP if possible.

Response: In response to the reviewer's comment, we have added relevant data and adjusted our statement.

Now, the evidence that *OsMADS17* regulates the transcription of *SDT* (*MIR156j*) includes:

(1) *SDT* has regional significant eGWAS signals in *OsMADS17* (Fig. 7c), which is a kind of genetic evidence in support of the regulation of *SDT* by *OsMADS17*.

(2) We validated this regulatory mechanism through transient transcriptional activity assay (Fig. 7de).

(3) We added another molecular assay, EMSA (Supplementary Fig. 16a), which further validated that *OsMADS17* directly binding to the 4.2-4.4 kb dOCR upstream of *SDT* and thus regulates *SDT* transcription.

(4) We also complemented a qRT-PCR analysis, which showed that *SDT* expression was moderately down-regulated in the young panicle of the *CR-Osmads17* mutant compared to WT (Supplementary Fig. 16b).

We believe that these pieces of evidence are sufficient to support that *OsMADS17* regulates *SDT* expression. However, as the regulatory network we constructed showed that *OsMADS17* may also regulate SPP by regulating other genes (Fig. 7j), we revised the title of the Figure7 from "*OsMADS17* affects SPP by regulating *SDT* transcription" to "*OsMADS17* regulates *SDT* transcription and affects SPP", and adjusted the relevant statements in the Abstract (line 35) and Introduction (line 80).

What is the meaning of the nodes circle size in Figure 7J?

Response: In Figure 7j, the size of each node circle indicates the number of nodes it connects to. Larger circles represent nodes with more connections, while smaller circles indicate nodes with fewer connections. We have added this information to the legend of Figure 7j.

5. The significant difference test should be added in the Supplemental Figure 15: Haplotype analysis of SDT and OsMADS17.

Response: Thank you for the suggestion. We have added the results of the significance difference tests to Supplementary Figure 15.

Reviewer #3 (Remarks to the Author):

Grain yield of rice is a quantitative trait which is mainly determined by tiller number, grain number and grain weight. In particular, the number of grains per panicle (panicle branching) is controlled by many quantitative trait loci (QTL), but it's also largely influenced by multiple environmental factors. In the past decades, several genes that control shoot apical meristem development and panicle branches have been identified and characterized by using GWAS, QTL analysis and map-based cloning approaches, however, the mechanisms underlying the interplay among these key regulators/genes still remain unclear. In this manuscript, the authors performed transcriptome-wide association analysis (TWAS) combined with GWAS and quantitative trait loci mapping, and identified new regulators of panicle branching and grain numbers. The authors also developed a method to identify the candidate genes and their regulatory works, and identified 36 putative new genes in regulating panicle branching. This new approach will help us to improve efficiency and maximize the genetic load of factors modulating quantitative traits in crops. The topic of this manuscript is interest for researchers follow trends and understanding of the genetic control of panicle architecture. However, there are deficiencies in experimental analysis that require further substantiation before the manuscript can be acceptable for publication.

1, previous studies by using integrating GWAS, eQTL, chromatin landscapes and/or gene expression have revealed the genetic control of key agronomic traits in many crops. Although many genes have been reported to have cis- (or trans-) variants that regulated

gene transcription, and thus causing variation of grain yield-related traits. To in-depth investigate the genetic basis and gene regulatory networks in the regulation of panicle architecture, the authors should combine with looking for epigenetic DNA methylation and chromatin landscapes relative to genome-wide transcription analysis of 275 rice varieties.

Response: We thank the reviewer for the insightful comments and suggestions. We agree that integrating GWAS, eQTL, and DNA methylation or chromatin landscape can further elucidate the genetic and molecular regulatory mechanisms of agronomic traits. We are planning to obtain DNA methylation or chromatin landscape data of young panicles in future studies to further investigate the genetic basis and gene regulatory network of rice panicle architecture.

2, according to the GWAS results of panicle architecture, the authors suggested that variations in panicle branching and grain number appear to be mainly regulated by many small-effect loci. In fact, unlike thousand-grain weight, panicle branching and grain number are largely affected by multiple environmental factors. I don't know if the TWAS results could be repeated from the same population which were planted in the different fields or geographic areas (or different years).

Response: We conducted TWASs on phenotypes from different geographical areas and years and compared the results. As shown in the Response Figs. 2-4, the TWAS results from different geographical areas and years are generally highly consistent (Response Figs. 2-4).

Response Fig. 2: Comparison of TWAS results for phenotypes of SPP from different geographical locations and years. The x - and y -axis values are the TWAS Z-scores for each gene in each year (or location). The histograms on the diagonal show the distribution of TWAS Z-scores for phenotypes in each year (or location). WH2016, WH2013, WH2014, and HN2013 indicate the field trials performed in Wuhan in 2016, Wuhan in 2013, Wuhan in 2014, and Hainan in 2013, respectively.

Response Fig. 3: Comparison of TWAS results for phenotypes of NPB from different geographical locations and years, as in Response Fig. 2.

Response Fig. 4: Comparison of TWAS results for phenotypes of PL from different geographical locations and years, as in Response Fig. 2.

3, in Fig.7, the authors showed that *OsMADS17* affected panicle architecture through regulating *miR156j* transcription. The authors only showed the panicle phenotype of the *osmads17* mutant, they did not present the evidence for supporting the genetic interaction between *OsMADS17* and *miR56j*.

Response: We thank the reviewer for this insightful comment and have responded to a similar concern raised by reviewer #2 (comment 4). Briefly, in the previous manuscript, we presented that *SDT* has regional significant eGWAS signals in *OsMADS17* (Fig. 7c), which is a kind of genetic evidence for the regulation of *SDT* by *OsMADS17*. And we validated this regulatory mechanism through transient transcriptional activity assay (Fig. 7de). In this revision, we added EMSA (Supplementary Fig. 16a), further confirming that *OsMADS17* directly binds to the 4.2-4.4 kb dOCR upstream of *SDT*. Additionally, we performed a qRT-PCR analysis, showing that *SDT* expression was moderately down-regulated in the young panicle of the *CR-Osmads17* mutant compared to WT (Supplementary Fig. 16b). We believe that this evidence is sufficient to support that *OsMADS17* regulates *SDT* expression. However, as the regulatory network we constructed showed that *OsMADS17* may also regulate SPP through other genes (Fig. 7j), we modified the statement accordingly. We have changed the title of the Figure7 from "*OsMADS17* affects SPP by regulating *SDT* transcription" to "*OsMADS17* regulates *SDT* transcription and affects SPP", and adjusted the relevant content in the Abstract (line 35) and Introduction (line 80).

4, the reference of *SDT* (*miR156j*) should be cited in the text

Response: We have added citations to the reference of *SDT* in line 171 and line 436 of the manuscript.

REVIEWERS' COMMENTS

Reviewer #2 (Remarks to the Author):

Thanks for the additional data and explanations. I have no other comments.

Reviewer #3 (Remarks to the Author):

In this revised version of the manuscript, the authors have added additional data and answered the question which I addressed, now, it is acceptable for publication.

Reviewer #2 (Remarks to the Author):

Thanks for the additional data and explanations. I have no other comments.

Response: Thank you for your review, we appreciate your support.

Reviewer #3 (Remarks to the Author):

In this revised version of the manuscript, the authors have added additional data and answered the question which I addressed, now, it is acceptable for publication.

Response: Thank you for your review, we are glad that you find the revised manuscript acceptable.